# An Isotonic Model of Neuron Swelling Based on Co-Transport of Salt and Water

**DOI:** 10.3390/membranes13020206

**Published:** 2023-02-07

**Authors:** Reinoud Maex

**Affiliations:** Biocomputation Research Group, School of Physics, Engineering and Computer Science, University of Hertfordshire, College Lane, Hatfield AL10 9AB, UK; r.maex1@herts.ac.uk

**Keywords:** brain, extracellular space, ions, concentration gradient, energy, mixing, osmosis, ischemia, co-transporter, KCC2

## Abstract

Neurons spend most of their energy building ion gradients across the cell membrane. During energy deprivation the neurons swell, and the concomitant mixing of their ions is commonly assumed to lead toward a Donnan equilibrium, at which the concentration gradients of all permeant ion species have the same Nernst potential. This Donnan equilibrium, however, is not isotonic, as the total concentration of solute will be greater inside than outside the neurons. The present theoretical paper, in contrast, proposes that neurons follow a path along which they swell quasi-isotonically by co-transporting water and ions. The final neuronal volume on the path is taken that at which the concentration of impermeant anions in the shrinking extracellular space equals that inside the swelling neurons. At this final state, which is also a Donnan equilibrium, all permeant ions can mix completely, and their Nernst potentials vanish. This final state is isotonic and electro-neutral, as are all intermediate states along this path. The path is in principle reversible, and maximizes the work of mixing.

## 1. Introduction

Neurons spend more than half of their energy building and maintaining ionic concentration gradients across their cell membrane [1]. Indeed, the distribution of monovalent ions, depicted as the physiological state in Table 1, is thermodynamically unstable, and requires the activity of an enzyme, the electrogenic 3${\mathrm{Na}}^{+}$/2$K^{+}$ pump, which hydrolyses adenosine triphosphate (ATP) to provide the energy for the expulsion of three Na${}^{+}$ ions against the entry of two K${}^{+}$ ions during a single pump cycle [2,3]. This electrogenic Na${}^{+}$/K${}^{+}$ pump explains not only the concentration gradients of the Na${}^{+}$ and K${}^{+}$ ions, but, because of the obligatory electro-neutrality of the bulk solutions, it also explains that of the monovalent anions such as Cl${}^{-}$ and HCO${}_{3}^{-}$ [4]. The physiological state is isotonic as the total solute concentration equals 304 mM on both sides of the neuronal plasma membrane. Please note that the divalent cations are discarded in the present study: they require a separate pump, their physiological concentrations are too low (<2 mM) to be of major influence in the present energy calculations, and blocking the Ca${}^{2+}$ channels does not prevent the neurons from swelling [5].

It could be said that all subsequent electrical activity of the neurons is caused by the mixing of ions between the intra- and extracellular compartments. This mixing follows a path that ultimately lowers the Gibbs energy [6]. During mixing the ions can do work, for instance by providing the driving force for the secondary transport of other ions, neuro-transmitters, or nutrients such as glucose [7].

Hyperactivity of the neurons, or energy deprivation, causes extracellular space to shrink [8]. Although this shrinkage has largely been attributed to the swelling of glial cells [9,10], neurons swell in a similar manner [5,11], despite their lack of functional aquaporins [12]. During severe ischemia, extreme swelling of the neurons (and glial cells) can reduce the volume fraction α taken by extracellular space from its physiological value of about 0.25 to less than 0.05 [13,14]. In its early stages, the process can be reversible, and a spontaneous transient form is known as spreading depolarization [15]. The concomitant mixing of the ${\mathrm{Na}}^{+}$, $K^{+}$ and ${\mathrm{Cl}}^{-}$ ions between the extra- and intracellular compartments is thought to accumulate solute within the neurons, causing their swelling [16]. Nevertheless, the mechanism of their swelling must be AQP4-independent [17]. More restricted volume changes (α decreasing from 0.23 to 0.14) have been observed between physiological sleep and wakefulness [18].

**Table 1 membranes-13-00206-t001:** Extra- and intraneuronal ion concentrations before and after mixing.

Ion Species	Physiological (α = 0.25) ^a^		Full Equilibrium (ω=0.024)	Transport
	Outside $K_{o}^{*}$ (mM)	Inside $K_{i}^{*}$ (mM)	Outside = Inside $K_{e}$ (mM)	Outside → Inside $c_{K}$ (mM)
${\mathrm{Na}}^{+}$	148	12	46	159.0
$K^{+}$	4	140	106	−7.0
${\mathrm{Cl}}^{-}$	113	8	34.25	121.5
${\mathrm{HCO}}_{3}^{-}$	29	10	14.75	30.5
$A^{-}$ ^b^	10 ^c^	134	103	0
Total	304	304	304	304

^a^ Adapted from Figure 3 of Reference [19]; ^b^ Impermeant anions, mostly negatively charged metabolites and
macromolecules; ^c^ See Reference [20].

In a previous theoretical study [4], it was suggested that the observed volume changes enhance the work done by the mixing of ions. Suppose for instance that the high-concentration compartment of an ion shrinks during the mixing process (in the present case the extracellular compartment for sodium or chloride), then this shrinkage will increase the ion’s concentration, whereas the simultaneous expansion of its low-concentration compartment lowers its concentration therein. Taken together, the concentration gradient is maintained at a higher level throughout the mixing process, enhancing the work that can be done by mixing of the ion.

The full picture is more complex, however, for two reasons. First, for other ions the high-concentration compartment may expand (for intracellular K${}^{+}$ for instance), reducing their work of mixing. Second, not all diffusible ions are miscible. About 40% of the intracellular ions are membrane-impermeant anions, mostly negatively charged amino-acids [21] or metabolites made negatively charged through phosphorylation [22]. The confinement of these anions to the intracellular compartment also prevents the complete mixing of the permeant ion species, as the equalization of the concentrations of the latter would leave the intracellular compartment negatively charged, violating the obligatory electro-neutrality of the bulk solutions.

This latter problem is usually dealt with by assuming that mixing evolves toward a Donnan equilibrium [16,23,24], at which the concentrations of all permeant ion species are distributed such that their Nernst potentials coincide (the so-called Donnan potential) (Table 2). The Donnan equilibrium is electro-neutral, but not isotonic, as the total solute concentration will be greater inside then outside the neurons (by 171 mM in Table 2). This osmotic imbalance of the Donnan equilibrium may be sufficient for other cell types to swell, but neurons, lacking functional aquaporins in their cell membrane [11,25], have been reported to withstand greater osmotic stresses than glial cells do [11,12] (but see [17]). It must also be borne in mind that a 1 mM difference in total solute concentration across a semi-permeable membrane corresponds to a hydrostatic pressure of 2576 N m${}^{-2}$ or 19.3 mm Hg, which is greater than the physiological intracranial pressure [26].

Instead of using aquaporins (and the default permeability of lipid-bilayer membranes for water molecules [27,28]), neurons transport water molecules across their cell membrane in an almost stoichiometric ratio with other solutes [11,20,25,28,29]. For instance, H${}_{2}$O is co-transported with K${}^{+}$ and Cl${}^{-}$ in the KCC2 co-transporter [28], which neurons strongly express [25,30,31], and other co-transporters have been shown to act as H${}_{2}$O channels, such as those for glutamate (EEA3-4) and lactate (MCT2) [10,11,29]. The H${}_{2}$O transport is driven by the combined chemical potentials of the ions (or protons) and H${}_{2}$O. Consequently, it does not need an osmotic imbalance, and can even run against an osmotic gradient [28,32]. The direction of transport can also reverse so that, under elevated extracellular K${}^{+}$ concentrations, H${}_{2}$O is transported into the neurons [10,33].

The present paper describes a path for neuronal swelling based on the co-transport of H${}_{2}$O and ions. The path preserves isotonicity and electro-neutrality, is in principle reversible, and maximizes the work of mixing. The model is applied to cytotoxic oedema, the early stage of neuron swelling during ischemia [34,35,36].

## 2. Materials and Methods

Parts of the model have been presented in a different form in the paper by Maex [4]. The analytical model consists of two compartments, one representing extracellular (or interstitial) space, the other the neuronal (or intracellular) volume (Figure 1). Please note that there is no separate compartment for glial cells; as explained in Section 4, arguments can be forwarded to allocate them to either compartment.

The two compartments are separated by a movable semi-permeable membrane (thick vertical line in Figure 1). During mixing the total volume *V*, and the number of particles of each ion species, are conserved.

With only the four monovalent ions of Table 1 being considered, the model has five thermodynamic state variables: the volume $V_{o}$ taken by extracellular space and the extracellular concentration $K_{o}$ of each of the four monovalent permeant ion species. The values of the corresponding intracellular quantities are determined by the conservation of the total volume *V* (always taken to be unity) and of the numbers of particles.

The symbol *K* will be used as shorthand for any of the four permeant monovalent ion species, with $K^{*}$ and $K^{\circ }$ denoting their initial and final concentration, and $K_{o}$ and $K_{i}$ that in the outer (extracellular or interstitial) and inner (neuronal) compartment. In the formulas, α always denotes the initial (physiological) volume fraction taken by extracellular space (α= 0.25 in the present study), hence $V_{o}^{*}=\alpha V$. The volume fraction taken by the extracellular compartment after mixing is denoted by ω and usually expressed as a fraction *w* of α (ω=wα). Hence $V_{o}^{\circ }=\omega V=w\alpha V=wV_{o}^{*}$, and likewise, $V_{i}^{\circ }=\left(1-\omega \right)V=\left(1-w\alpha \right)V=yV_{i}^{*}$, where the relative volume change *y* of the inner compartment is given by y=(1−wα)/(1−α).

The change in concentration of an ion species during mixing will depend on the change in volume of the host compartment on the one hand, and on the loss or gain of particles by transport across the cell membrane on the other hand. If the electrolytes are ideal solutions, then the change in compartmental volume must equal the volume of solvent (water) that is transported through the membrane, and both causal factors can be captured in a single parameter: the ratio of solvent to solute transport. This ratio is taken constant throughout the mixing process, with value $a_{K}$ for ion species *K*. More particularly, the amount ΔK (in moles) transported of ion species *K* holds a constant relationship to the volume ΔV of transported solvent (in m${}^{3}$)
(1)$$\Delta V=a_{K}\Delta K.$$

As ΔV must also represent the change in volume of the recipient compartment, it is more convenient to rewrite Equation (Equation 1) as
(2)$$c_{K}\Delta V_{o}=\Delta K_{o},$$
where $\Delta V_{o}$ denotes the change in volume of the extracellular compartment and $\Delta K_{o}$ is the amount of solute *K* (in number of moles) added to (or lost by) the same compartment. Parameter $c_{K}=a_{K}^{-1}$ denotes the constant concentration at which ion species *K* is co-transported with the solvent throughout the mixing process. The value of $c_{K}$ is positive when solute and solvent are transported in the same direction, negative when their transport takes opposite directions. For the impermeant anions, $c_{A^{-}}=0$. Please note that $c_{K}$ must not be interpreted to be the concentration of solute within the co-transporter channel; rather it should be considered to be the ratio of the molar amount of transported ions of species *K* to the total amount of solvent transported across all water channels.

The value of $a_{K}$ (or $c_{K}$) can be calculated for each ion species *K* from the initial and final conditions. Taking a change in volume $\Delta V_{o}=w\alpha -\alpha$, and an amount of transported solute $\Delta K_{o}=w\alpha K_{o}^{\circ }-\alpha K_{o}^{*}$, it follows from Equation (Equation 1) that
(3)$$\begin{array}{c} a_{K}=\frac{1-w}{K_{o}^{*}-wK_{o}^{\circ }}, \end{array}$$
or
(4a)$$\begin{array}{cc} c_{K} & =\frac{K_{o}^{*}-wK_{o}^{\circ }}{1-w}\\ \; \end{array}$$
(4b)$$\begin{array}{cc} \; & =\frac{K_{i}^{*}-yK_{i}^{\circ }}{1-y}, \end{array}$$
where Equations ([Disp-formula FD4a-membranes-13-00206]) and (4b) are warranted to yield the same value of $c_{K}$ by the conservation of the number of particles.

As noted above, mixing rule Equation (Equation 1) is consistent with the observation that pure water transport through the neuronal membrane is largely impeded by a lack of functional aquaporins [12], and that, instead, water molecules can be co-transported in an almost fixed stoichiometric ratio with solute particles [11,17,20,25,28,29,32].

## 3. Results

### 3.1. Construction of the Isotonic Path

The volume fraction ω taken by extracellular space at the state of full equilibrium in Table 1 was calculated as the volume at which the (unspecified) impermeant anions $A^{-}$ acquire the same concentration outside and inside the neurons. If extracellular space shrinks from its physiological volume fraction α to a final volume ω=wα, then the concentrations vary as
(5a)$$\begin{array}{cc} {}[A^{-}{]}_{o} & =\frac{1}{w}[A^{-}{]}_{o}^{*}\\ \; \end{array}$$
(5b)$$\begin{array}{cc} {}[A^{-}{]}_{i} & =\frac{1-\alpha }{1-w\alpha }[A^{-}{]}_{i}^{*}=\frac{1}{y}[A^{-}{]}_{i}^{*}, \end{array}$$
where *w* and *y* are the relative volume change of the extra- and intra-cellular compartment, respectively. The state of equilibrium in Table 1, at which ${\left[A^{-}\right]}_{e}={\left[A^{-}\right]}_{o}={\left[A^{-}\right]}_{i}$ = 103 mM, is reached when extracellular space has shrunken to 9.7% of its initial volume ($w_{e}=0.0971$) so that it occupies only 2.4% of the entire volume of brain tissue (ω = 0.0243). The neuronal compartment has expanded by 30% (*y* = 1.3).

At these volumes, all permeant ion species are also able to equilibrate completely, with the equilibrium concentration $K_{e}$ of permeant ion species *K* measuring
(6)$$K_{e}=K_{o}^{\circ }=K_{i}^{\circ }=\alpha K_{o}^{*}+\left(1-\alpha \right)K_{i}^{*}.$$
Please note that the state of full equilibrium in Table 1 is only used for the construction of the isotonic path; it is not suggested that swelling neurons ever reach that state in their life-time.

Substituting the values of $w_{e}$ and $K_{e}$ for *w* and $K_{o}^{\circ }$ in Equation ([Disp-formula FD4a-membranes-13-00206]) yields for the constant concentration at which permeant ion *K* is transported across the membrane
(7)$$\begin{array}{cc} c_{K} & =\frac{K_{o}^{*}-wK_{e}}{1-w}, \end{array}$$
with $K_{o}^{*}$ denoting the initial concentration as given in the first column of Table 1. The value of $c_{K}$ for each permeant ion species is listed in the last column, with the negative sign for $c_{K^{+}}$ indicating that the net transport of $K^{+}$ ions (outward) is against the direction of water flow (inward). It is also clear from Table 1 that the total membrane transport is electro-neutral, and isotonic with the initial total solute concentration of 304 mM.

In Appendix A, two theorems are formally proven to the effect that when the initial and final states are electro-neutral and isotonic, and the transport of ions follows Equation ([Disp-formula FD4a-membranes-13-00206]) and is also electro-neutral and isotonic, then the entire path with all intermediate states is electro-neutral and isotonic as well. Hence all intermediate states along the isotonic path connecting the physiological state and the state of full equilibrium in Table 1 are electro-neutral and isotonic.

Evidently, these transport concentrations, as given by the values of $c_{K}$, depict only the net transport of each ion species without indication of the mechanism or channel involved. For instance, the weak net outward transport of K${}^{+}$ ($c_{K^{+}}$ = −7 mM) does not imply that the K${}^{+}$ ions are little involved in the mixing process, even though the rise of their extracellular concentration is accounted for almost completely by the contraction of extracellular space (see Section 3.3). Instead, many $K^{+}$ ions may first have been expelled (for instance through Kv channels) before re-entering the neuron together with ${\mathrm{Cl}}^{-}$ by reverse flow through the KCC2 co-transporter [10,33,37], which acts as a H${}_{2}$O channel [11,20,28,29,32].

For the total membrane transport to be isotonic, about 183 H${}_{2}$O molecules should be transported with each ion (55.6 M/0.304 M). If *all* H${}_{2}$O were to enter the neuron through the KCC2 co-transporter, then each Cl${}^{-}$ ion should be accompanied by 183×304/121.5=458 H${}_{2}$O molecules, given that $c_{{\mathrm{Cl}}^{-}}$ = 121.5 mM. This number of H${}_{2}$O molecules could easily be accommodated in the KCC2 co-transporter, which is thought to transport about 500 H${}_{2}$O molecules for each Cl${}^{-}$ ion [28,32]. Even so, not all H${}_{2}$O molecules need to enter the neuron through co-transporters, as lipid bilayers have an intrinsic ion-independent H${}_{2}$O permeability [27], which accounted for one third of the H${}_{2}$O transport in choroid epithelial cells [28].

### 3.2. Qualitative Comparison of the Isotonic Path and the Donnan Equilibria

Figure 2 illustrates the major differences between the proposed isotonic path (black curves) and the Donnan equilibria calculated at the intermediate volumes $\tilde{\alpha }$ (red curves). The initial, physiological state is depicted by the values of the black curves at $\tilde{\alpha }$ = 0.25. At the state of full equilibrium of Table 1 ($\tilde{\alpha }=\omega =0.024$) the black and red curves intersect. As argued above, on the isotonic path the total solute concentration both inside and outside the neurons remains clamped at the initial value of 304 mM (Figure 2A), whereas the Donnan equilibrium is isotonic only at $\tilde{\alpha }=0.024$. On the other hand, the Nernst potentials of the different ion species only gradually converge to a value of zero mV at $\tilde{\alpha }=0.024$ along the isotonic path, whereas they all coincide at the Donnan potential corresponding to each intermediate Donnan equilibrium (Figure 2B).

For the isotonic path to be reversible, all intermediate states must be near thermodynamic equilibrium. It must therefore be assumed that, at each intermediate volume $\tilde{\alpha }$, the chemical potentials of the permeant ions are balanced by an equivalent chemical potential of the energy-providing ATP, whose concentration then decreases during the mixing process. In other words, at each volume $\tilde{\alpha }$ the cytosolic concentration of ATP and its metabolites must be such that the Gibbs energy of the hydrolysis of ATP equals the work required for the reversible and stoichiometric transfer of Na${}^{+}$ and K${}^{+}$ ions (and of the obligatory anions needed to conserve electro-neutrality) [3]. Please also note that all mixing is purely entropy-driven in the present model, and that, as argued in Section 4, the membrane potential has been neglected in the calculations, as this quantity, being itself a consequence of the ion gradients, represents only a tiny fraction of the energy stored in them [4]. Although the membrane potential can determine the speed of the time-course to equilibrium, it cannot affect the equilibrium itself [38]. For all processes to be reversible, it must further be assumed that heat produced by exothermal reactions (for instance channel transport discharging the membrane capacitor) is recovered by endothermal processes, as suggested by El Hady and Machta [39].

In contrast, since the Donnan equilibrium is electrochemically stable, ATP can be assumed to have zero potential throughout for the red curves in Figure 2. For the swelling toward a Donnan equilibrium to be physically plausible and reversible, on the other hand, the osmotic pressure gradient must be balanced, for instance, by elastic potential energy stored in the membrane and cytoskeleton [40].

### 3.3. Variation of Ion Concentration with Extracellular Volume

It can be shown [4] that, when mixing obeys Equation (Equation 1), the concentrations $K_{o}$ and $K_{i}$ of ion species *K* are hyperbolic functions of the volumes $V_{o}$ and $V_{i}$ of the host compartments
(8a)$$V_{o}=\frac{C}{1-a_{K}K_{o}}$$
(8b)$$V_{i}=\frac{D}{1-a_{K}K_{i}},$$
where the integration constants *C* and *D* are determined by the initial conditions
(9a)$$\begin{array}{cc} C & =V_{o}^{*}\left(1-a_{K}K_{o}^{*}\right) \end{array}$$
(9b)$$\begin{array}{cc} D & =V_{i}^{*}\left(1-a_{K}K_{i}^{*}\right), \end{array}$$
so that
(10a)$$\begin{array}{cc} K_{o} & =c_{K}-\frac{c_{K}-K_{o}^{*}}{w} \end{array}$$
(10b)$$\begin{array}{cc} K_{i} & =c_{K}-\frac{c_{K}-K_{i}^{*}}{y}. \end{array}$$
During swelling of the neurons $V_{i}$ increases (y> 1), whereas the extra-cellular volume $V_{o}$ decreases (w< 1).

Figure 3A,C plots the concentration-volume functions given by Equations ([Disp-formula FD10a-membranes-13-00206]) and (10b), using for each ion species the values of $c_{K}$ and $K^{*}$ given in Table 1. For instance, [K${}^{+}{]}_{o}$ increases during skrinkage of extracellular space (when $\tilde{\alpha }$ decreases in Figure 3), and this increase is enhanced by a small outward flux of the K${}^{+}$ ions ($c_{K^{+}}=-7$ mM). In contrast, [Na${}^{+}$]${}_{o}$ and [Cl${}^{-}$]${}_{o}$ drop because the Na${}^{+}$ and Cl${}^{-}$ ions flow at a high concentration into the neurons ($c_{{\mathrm{Na}}^{+}}=159$ mM, $c_{{\mathrm{Cl}}^{-}}=121.5$ mM, see Table 1). This drop becomes most prominent once [K${}^{+}$]${}_{o}$ exceeds about 10 mM (Figure 3C), in agreement with experimental reports [41,42]. No physical solutions exist on the isotonic path for $\tilde{\alpha }<0.02$ because [Na${}^{+}{]}_{o}$ would become negative.

The concentration of the impermeant anions, [A${}^{-}{]}_{o}$, follows approximately the course of [K${}^{+}{]}_{o}$ and can increase by a factor greater than 10 (not shown). Hence the observed apparent ’anion gap’ (the difference between the concentrations of the permeant cations and anions) or apparent decrease of the osmolarity of the extracellular solution [43], can be accounted for by the rise in the concentration of the impermeant anions during the shrinkage of extracellular space.

Figure 3B,D plots, over the same volume range, the ionic concentrations at Donnan equilibrium. In contrast to their coordinated decrease on the isotonic path, [Na${}^{+}{]}_{o}$ and [Cl${}^{-}$]${}_{o}$ diverge at the smallest extracellular volumes in Figure 3B.

**Figure 3 membranes-13-00206-f003:**
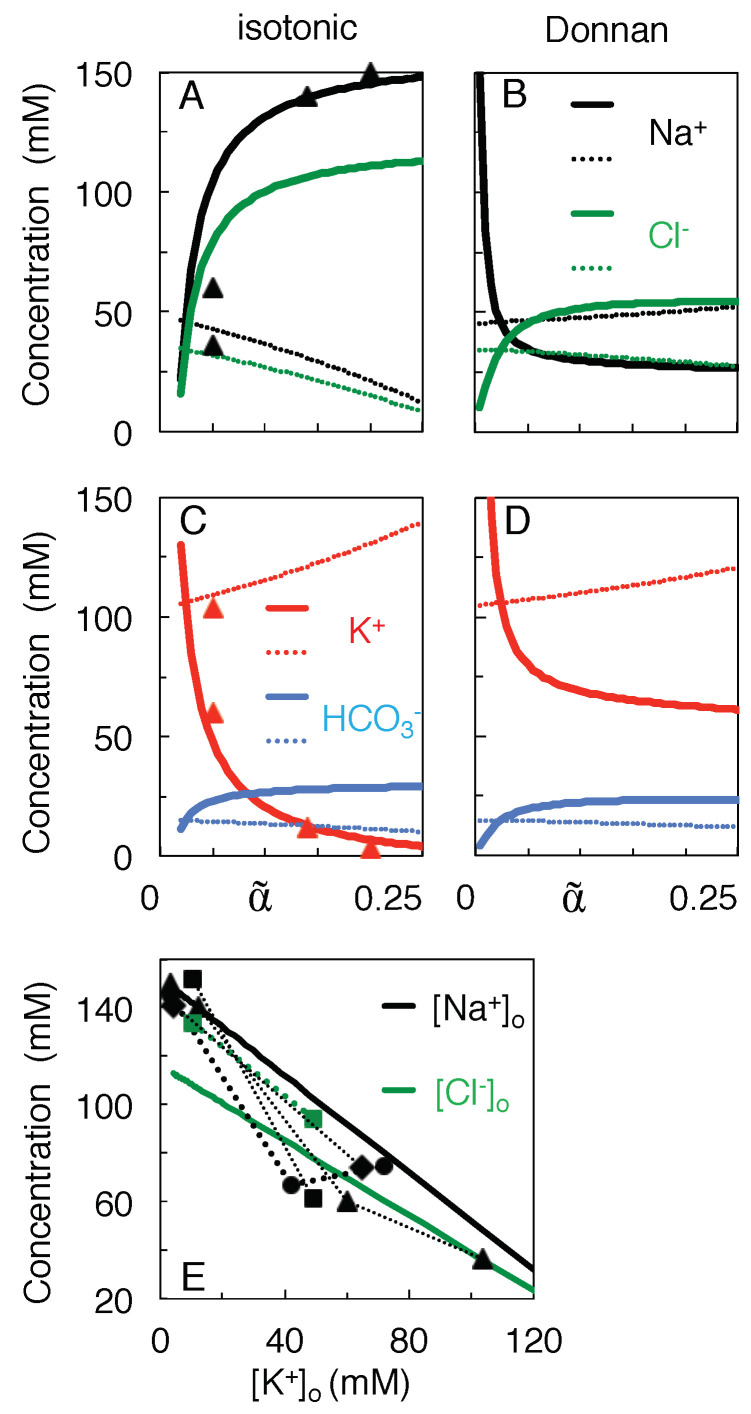
Concentration-volume curves for the permeant ions during isotonic mixing (**A**,**C**), as compared to mixing to the Donnan equilibrium at each intermediate volume $\tilde{\alpha }$ (**B**,**D**). Thick and thin curves plot extra- and intracellular concentrations, respectively. Data points △ in (**A**,**C**) are from Supplementary Table S2 of Reference [44]. (**E**) Theoretical prediction of the profile of $[{\mathrm{Na}}^{+}{]}_{o}$ (black solid line) and $[{\mathrm{Cl}}^{-}{]}_{o}$ (green) against $[K^{+}{]}_{o}$. Data points are from Reference [44] (△), [45] (□), [46] (∘), and [9] (⋄). Data from the same reference are connected by dotted lines, but may have been compiled from different experiments.

Finally, Figure 3E combines the concentration-volume profiles from (A) and (C) to plot curves of [Na${}^{+}{]}_{o}$ and [Cl${}^{-}{]}_{o}$ against [K${}^{+}{]}_{o}$ for the isotonic model. It can be derived from Equation ([Disp-formula FD10a-membranes-13-00206]) that these curves are straight lines of slope minus the ratio of the physiological concentration gradients ($K_{o}^{*}-K_{i}^{*}$) (hence −1 for Na${}^{+}$, and −0.77 for Cl${}^{-}$). Please note that of the experimental data points, those of Mori et al. [9] were measured at 5 min after the induction of ischemia, those of Windmüller et al. [45] and Stiefel and Marmarou [46] after 15 min or more.

### 3.4. Work of Mixing versus Osmotic Work

The ionic concentration gradients constitute a major store of energy for neurons. In a previous article [4], it was suggested that more work would be done by mixing if the neurons swelled, and interstitial space shrank, during the mixing process. In calculating the work of mixing, it is assumed that the mixing is carried out under reversible conditions, hence that all energy can be retrieved as work.

With the above convention, the maximum work done, or Gibbs energy retrieved, by the passive transport of solute *K*, can be expressed as
(11)$$W=\Delta G=-\mathrm{RT}\int_{K_{o}^{*}}^{K_{o}^{\circ }}\ln\frac{K_{o}}{K_{i}}\mathrm{dK}=\mathrm{RT}\int_{K_{o}^{\circ }}^{K_{o}^{*}}\ln\frac{K_{o}}{K_{i}}d\left(K_{o}V_{o}\right),$$
where *R* is the universal gas constant (8.3145 J mol${}^{-1}$ K${}^{-1}$), and *T* the physiological temperature (310 K).

Equation (Equation 11) can be solved as (see Maex [4])
(12)$$\begin{array}{ccc} \frac{\Delta G}{\mathrm{RT}} & = & V_{o}^{*}K_{o}^{*}\ln K_{o}^{*}-wV_{o}^{*}K_{o}^{\circ }\ln K_{o}^{\circ }\\ & & +V_{i}^{*}K_{i}^{*}\ln K_{i}^{*}-yV_{i}^{*}K_{i}^{\circ }\ln K_{i}^{\circ }\\ & & +\frac{C}{a_{K}}\ln w\\ & & +\frac{D}{a_{K}}\ln y, \end{array}$$
where *C* and *D* are given by Equations ([Disp-formula FD9a-membranes-13-00206]) and (9b), and *w* and *y* denote for each compartment the relative change of its volume (as in Equations ([Disp-formula FD5a-membranes-13-00206]) and (5b)).

Equation (Equation 12) can be given a straightforward physical interpretation. The first two terms represent the change in energy of solute *K*, after mixing, in the extracellular compartment, and the third and fourth terms the corresponding change in the intracellular compartment. The fifth and sixth terms can be shown to represent the osmotic work done by solute *K* in displacing the membrane barrier (making the neurons swell).

Indeed, if the solutions are ideal so that the gas laws apply, then the work done by the extracellular ions of species *K* in changing the volume of the extracellular compartment from $V_{o}^{*}$ to $V_{o}^{\circ }$ can be calculated as
(13a)$$\begin{array}{cc} \mathrm{RT}\int_{V_{o}^{*}}^{V_{o}^{\circ }}K_{o}dV_{o} & =\mathrm{RT}\int_{\alpha V}^{w\alpha V}\frac{V_{o}-C}{a_{K}V_{o}}dV_{o}\\ \; & =\mathrm{RT}\left(-\frac{\alpha (1-w)V}{a_{K}}-\left[\frac{C}{a_{K}}\ln w\right]\right), \end{array}$$
where Equation ([Disp-formula FD9a-membranes-13-00206]) has been used for *C* in the integrand. In the same manner, for the work done by the intracellular ions of species *K*, using Equation (9b),
(13b)$$\begin{array}{cc} \mathrm{RT}\int_{V_{i}^{*}}^{V_{i}^{\circ }}K_{i}dV_{i} & =\mathrm{RT}\int_{(1-\alpha )V}^{(1-w\alpha )V}\frac{V_{i}-D}{a_{K}V_{i}}dV_{i}\\ \; & =\mathrm{RT}\left(\frac{\alpha (1-w)V}{a_{K}}-\left[\frac{D}{a_{K}}\ln y\right]\right). \end{array}$$

Hence, the net work done by the mixing of solute *K* across the cell membrane, as calculated by Equation (Equation 12), equals the difference in chemical energy of ion species *K* before and after the mixing (the first four terms in Equation (Equation 12)), minus the osmotic work done by *K* in changing the volume (Equations ([Disp-formula FD13a-membranes-13-00206]) and ([Disp-formula FD13b-membranes-13-00206])), or
$$\begin{array}{cc} \mathrm{work}\;\mathrm{done}\;\mathrm{by}\;\mathrm{mixing} & =\mathrm{decrease}\;\mathrm{in}\;\mathrm{chemical}\;\mathrm{potential}\;\mathrm{energy}\\ \; & -\mathrm{osmotic}\;\mathrm{work}\;\mathrm{done}. \end{array}$$

For mixing at constant volume (w=y=1) the fifth and sixth terms of Equation (Equation 12) vanish, whereas for purely volume-induced concentration changes without transport of solute ($c_{K}={\left(a_{K}\right)}^{-1}=0$, as is the case for the impermeant anions $A^{-}$), the first four and the last two terms can be shown to cancel each other, and no work by mixing is done [4].

Figure 4A plots for each ion species the work of mixing done along the isotonic path starting from the physiological state at α = 0.25 down to each intermediate volume $\tilde{\alpha }$. Most of the work, as calculated from Equation (Equation 12), is done by the Na${}^{+}$ and Cl${}^{-}$ ions, in accordance with their larger values of $c_{K}$. Obviously the impermeant anions A${}^{-}$ can do only osmotic work, which is calculated as the sum of the right-hand sides of Equations ([Disp-formula FD13a-membranes-13-00206]) and ([Disp-formula FD13b-membranes-13-00206]) and is plotted by the orange curve in Figure 4B. The greater part of the work done by the K${}^{+}$ ions is also osmotic in nature (red curve), as it is their high-concentration compartment that swells. In contrast, Na${}^{+}$ and Cl${}^{-}$ *receive* osmotic work, as indicated by their negative energy values in Figure 4B (black and green curves), owing to the shrinkage of their high-concentration compartment. The work done on the Na${}^{+}$ and Cl${}^{-}$ ions causes their concentration gradients to be maintained at higher levels throughout the mixing process, and consequently enhances their work of mixing in Figure 4A. Overall, however, no osmotic work is done along the isotonic path (horizontal curve in Figure 4C), because the energy needed for water to enter the neurons is delivered by the concentration gradients of the co-transported ions, for instance in the KCC2 co-transporter [32].

To obtain the corresponding energies of mixing toward the Donnan equilibria (Figure 4D–F), separate values of $c_{K}$ had to be calculated for each intermediate volume $\tilde{\alpha }$ by substituting the value of $\tilde{\alpha }$/α and the concentration $K_{o}$ at Donnan equilibrium for *w* and $K_{o}^{\circ }$ in Equation ([Disp-formula FD4a-membranes-13-00206]). In this case, the total osmotic work vanishes only at $\tilde{\alpha }$ = α = 0.25 and at $\tilde{\alpha }=\omega$ = 0.024 (see Figure 4F), in the first case because the mixing is done at constant volume, in the second case because the Donnan equilibrium is identical to the state of full equilibrium in Table 1 and has actually been reached along the isotonic path. In between these states the osmotic work first rises when the neurons start swelling (Figure 4F), but decreases again when at smaller volumes $\tilde{\alpha }$ the osmotic imbalance diminishes, until it vanishes at the equilibrium point. When extracellular space shrinks even further, the overall osmotic work becomes negative as the polarity of the Donnan potential reverses (see Figure 2B). Mixing at constant volume $\tilde{\alpha }=\alpha =0.25$ to the Donnan equilibrium yields a work of mixing of 197 J/l (Figure 4F), equal to the amount of work done by mixing along the isotonic path when extracellular space shrinks by 54% (to $\tilde{\alpha }$ = 0.115 in Figure 4C).

Finally, the work of isotonic mixing to the state of full equilibrium (269.8 J/l at $\tilde{\alpha }=\omega$ = 0.024 in Figure 4C) must be, and is indeed, identical to the work that would be done by complete mixing at the *constant* volume of $\tilde{\alpha }=\alpha =0.25$ if all the impermeant anions were made permeant. In this case, all ions would reach the same final concentrations as in Table 1, only the final position of the membrane barrier would be different. This work done by complete isochoric mixing can be calculated from Equation (Equation 12) by setting w=y=1 and also allowing the $A^{-}$ to mix. The isochoric work is distributed among the ion species as: 80.3 (Na${}^{+}$), 66.8 (K${}^{+}$), 64.4 (Cl${}^{-}$), 53.1 (A${}^{-}$) and 5.1 J/l (HCO${}_{3}^{-}$). The corresponding values for mixing to full equilibrium along the isotonic path in Figure 4A are 138.5, 8.6, 109.4, 0.0, and 13.3 J/l, respectively. The comparison with isochoric mixing is only valid, of course, if the swelling neurons store only a negligible amount of elastic energy in their plasma membrane, for instance by using membrane buffers to rapidly adjust their surface area, as was observed to occur in hypo-osmotic stress experiments [47,48].

In conclusion, the isotonic path allows neurons (or cells in general) to recover the work of mixing that is lost owing to the impermeability of the cell membrane for charged metabolites. Although the impermeant anions cannot do work by mixing, the osmotic work they do in making the neurons swell raises the chemical potential of some of the permeant ions, allowing the latter (mostly extracellular Na${}^{+}$ and Cl${}^{-}$) to recover that work as work of mixing.

## 4. Discussion

In most theories of cytotoxic edema, neuron swelling is caused by the precipitous and irreversible release of the Gibbs energy stored in the ionic concentration gradients [35,44,49]. In most of these theories the swelling is also the consequence of an osmotic imbalance between the interior and exterior of the cells after the mixing (but some authors have argued that the swelling is non-osmotic in nature [36,49,50]).

The present study assumes that the ions, by mixing, can do work for the cell’s survival. Given that the relative volumes of the extra- and intracellular brain compartments vary on a diurnal basis [14,18], evolution may have optimized the mixing process [51], for instance by making it reversible (in the thermodynamic sense). Reversible mixing would maximize the amount of energy which neurons can retrieve (see Figure 5 for a mechanical analog). The predicted Na${}^{+}$ and K${}^{+}$ concentrations of Figure 3A,C reproduce the experimental data acceptably well, even without any attempt to adjust the model parameters. The energy profiles of Figure 4 confirm that cell swelling enhances the work of mixing.

Although the differences between the two kinds of model may become subtle in some implementations (see below), the therapeutic implications are profound, as in the present model the early stages of swelling could be regarded as an attempt by the neurons to retrieve more energy, and hence as a physiological process [18,51].

The model, being described using equilibrium thermodynamics, lacks a time-scale, but it should be restricted to processes that are slow enough for the chemical potentials to equilibrate (see Section 3.2), and fast enough so that the total volume and the numbers of particles can be considered constant. Given that the physiological ATP levels of the brain have been estimated to be sufficient to maintain metabolism for only one minute in ischemia [43], and that cerebrospinal fluid starts to infiltrate the brain parenchyma within a few minutes after cardiac arrest [35], the time span of the isotonic model should be set to a few minutes. For less severe anoxia without changes in the blood–brain barrier, or for physiological swelling [18], there is in principle no time limit on its validity. The latter also holds for experiments in which an external source of energy is applied through an artificial gradient, for instance between the immersed brain and an isotonic salt solution [35,38,49].

Before addressing the limitations, or missing components, of this simplified model, the importance of some of its features must be clarified.

(1)An isotonic path was chosen to conform to the observation that neurons lack aquaporins that can transduce osmotic forces [11,12,36], and isotonicity was preserved by the flow of water through co-transporters. Another, indistinguishable way of implementing isotonic swelling would be to have water permeate the lipid-bilayer membrane in response to each infinitesimal change in tonicity, using a model of coupled transport of water and solute [49,50].(2)The choice for the KCC2 co-transporter was motivated by its expression in neurons [30], and the repeated observation that swelling, or water transport, is strongly reduced or delayed in the absence of extracellular Cl${}^{-}$ ions [5,11,28,49]. Nevertheless, blockade of the KCC2 co-transporter reduced dendritic beading by only 36% in pyramidal cells [11], and Cl${}^{-}$ ions have also been observed to enter neurons through voltage- and ligand-gated channels [5].(3)Mixing was assumed to follow a path to the full equilibration of all ion species (Table 1), without requiring, however, that neurons travel the entire trajectory to equilibrium, a state they probably never reach in their lifetime.This state of full equilibrium is also the state of least energy (at constant entropy) or maximum entropy (at constant energy) [6], and thus being the state to which the system must evolve in the absence of energy supply, it allows components of the system to be ignored which can influence the speed of evolution but not its final state (such as the membrane potential, see below).Of course, other isotonic paths can be constructed, but they reduce the work of mixing. For instance, incrementing and decrementing $c_{{\mathrm{Na}}^{+}}$ and $c_{K^{+}}$ by 20 mM, respectively, to values of 179.0 and −27.0 mM, would preserve isotonicity and electro-neutrality but reduce the work of mixing from 269.8 to 263.2 J/l. Furthermore, the increased Na${}^{+}$ transport would deplete the extracellular Na${}^{+}$ store before the impermeant anions can equilibrate (more precisely, after shrinkage of extracellular space to a volume fraction of 0.05 as compared to 0.024 for mixing to full equilibrium).(4)Reversibility of the path requires that the direction of transport through the KCC2 co-transporter can be reversed [10,33,37]. Payne [33] reports that the driving force for K-Cl co-transport at physiological concentrations is close to thermodynamic equilibrium, the transport reversing from efflux to influx when $[K^{+}$]_*o*_ exceeds 5 mM. Backward operation of the Na${}^{+}$/K${}^{+}$ pump, on the other hand, has been described experimentally only under extreme ionic and ATP potentials [52,53,54]. Although it would provide a clear example of how ions might do work for the cell’s survival, recent experimental support for pump reversal is lacking, and the rate of the reverse reaction and the pump capacity may be too low for it to be of relevance in acute ischemic brain injury.

Inevitably, the present analytical model lacks anatomical and physiological detail. The non-modeled membrane potential is expected to have little influence on the energy calculations [4], but will of course determine the details of the path when more variables, such as the conductances of the ion channels, are included. The voltage-dependent opening and closing of channels may have two effects. First, it could temporarily and selectively enhance the transport of a particular ion species, in apparent disagreement with the present model where the $c_{K}$’s are considered constant. This constancy of the transport concentrations could still be fulfilled, however, on a more coarse-grained time-scale, in the same manner that fast voltage fluctuations such as an action potential should not be regarded as deviations from the principle of electro-neutrality of the bulk solutions. The second effect of a voltage-induced ion-specific increase in transport would be the selective accumulation of that ion in, or depletion from, one of the compartments. Neural death, for instance, is thought to be a consequence of Na${}^{+}$[5] or Ca${}^{2+}$ overload [55]. The only prediction the present isotonic model can make at this stage is that, on the time-scale of the swelling, the concentrations of all ion species should keep a linear relationship to each other, as indicated by the straight solid lines in Figure 3E. More precise predictions may require numerical models that explicitly implement the membrane potential and the expression of ion channels and transporters [38,56,57,58].

A missing component in a two-compartmental model are the glial cells, which occupy 5–15% of brain tissue [43,59]. Arguments can be forwarded to allocate them to either compartment. As scavengers of the K${}^{+}$ ions released by neurons, they could be allocated as K${}^{+}$ buffers to the extracellular compartment [4,57]. On the other hand, the re-uptake of K${}^{+}$ ions does not seem to be essential for their swelling, nor are their aquaporins (AQP4) [10]. Glial cells have been suggested to swell by co-transport of water and acid-base-regulating substances such as bi-carbonate (via NBCe1, the Na-HCO${}_{3}^{-}$ co-transporter type 1) or lactic acid (via the H${}^{+}$-coupled monocarboxylate transporter MCT4) [10]. In that sense, glial cells may also be considered to swell isotonically, and for that reason be allocated to the cellular (neuronal) compartment.

Another restriction of the present model is that the number of ions of each species is conserved. Consequently, it cannot be applied to buffering mechanisms that would act as sources or sinks of free ions in both compartments, such as the parallel decrease of [HCO${}_{3}^{-}$]_*o*_ and [HCO${}_{3}^{-}$]_*i*_ caused by the accumulation of CO${}_{2}$ during ischemia (often the bicarbonate concentrations reported in the literature have been derived indirectly from pH and pCO${}_{2}$ measurements using the Henderson-Hasselbach formula [43]). In addition, some metabolites may build up in the intracellular compartment during ischemia [43] or spreading depolarization [15], such as lactate and phosphates [15,43]. The equilibrium concentrations of these compounds may also change by differences in pH between the intra- and extracellular compartments [43,60]. Moreover, the breakdown of the primary concentration gradients will also affect the secondary gradients. Glutamate, for instance, the most abundant intracellular anion [21], may start permeating the membrane and leak from the cell through reverse flow in the Na${}^{+}$-driven glutamate transporter [61,62]. Calculating the effects of such compounds would require modeling brain pH homeostasis [63,64] and the pH-dependence of ATP hydrolysis [60,65].

This absence of ion buffering mechanisms in the model can also affect other (non-modeled) ion gradients. Ca${}^{2+}$ and Mg${}^{2+}$ are heavily buffered both outside and inside the neurons, and including their buffering may generate a considerable Ca${}^{2+}$ flow across the cell membrane (unpublished observations by the author). The Ca${}^{2+}$ ions may enter the ischemic neurons through missing components such as the activated NMDA receptors and Cav channels, or by reverse flow in the Na${}^{+}$/Ca${}^{2+}$ exchanger.

The conservation of particles and of the total volume also precludes the model from being used for the study of the ionic and vasogenic phases of brain edema [34,35]. Nevertheless, if during prolonged ischemia the permeability of the blood–brain-barrier for Na${}^{+}$ and Cl${}^{-}$ ions increases (or if other cation transporters are expressed [34]), then another two-compartmental model could be set up, this time between the blood plasma and the (almost equilibrated) brain parenchyma. Again, further swelling of the brain could be attributed to co-transport and/or osmotic forces, the latter enhanced, for instance, by the accumulation of autolytic products in the parenchyma [49].

Evidently, the absence of a vascular compartment also precludes the model from being used for the study of other factors involved in ischemic brain injury, such as the osmotic and hydrostatic pressure differences between intra- and extravascular space, the importance of perivascular space and the brain lymphatic system [35,66], or the paradoxical deterioration often observed after re-perfusion.

A final remark concerns the poorly specified impermeant anions in extracellular space, which may include proteins and polypeptides such as albumin and β-amyloid. Even though the extracellular (and extravascular) concentrations of these compounds are lower in the brain than in other tissues [26,67] because of the blood–brain barrier, the same blood–brain barrier may cause their concentrations to rise by a factor greater than ten during the shrinkage of extracellular space. Hence, although isotonic mixing may protect the neurons against osmotic stress, the increase in concentration of certain proteins may put these at risk of aggregation.

## Figures and Tables

**Figure 1 membranes-13-00206-f001:**
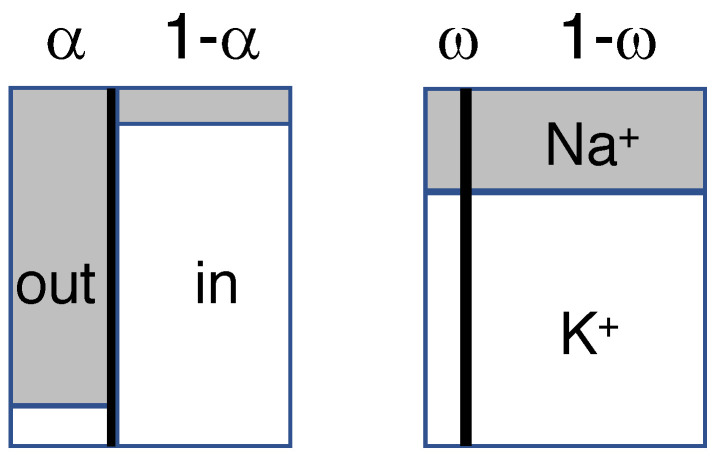
Diagram of the model, showing the extra- and intracellular compartments, separated by the membrane barrier (thick vertical line), before (**left**) and after mixing (**right**). Widths (not to scale) indicate relative compartmental sizes, heights relative concentrations of Na${}^{+}$ (gray) and K${}^{+}$ (white). Only the monovalent cations of Table 1 are shown.

**Figure 2 membranes-13-00206-f002:**
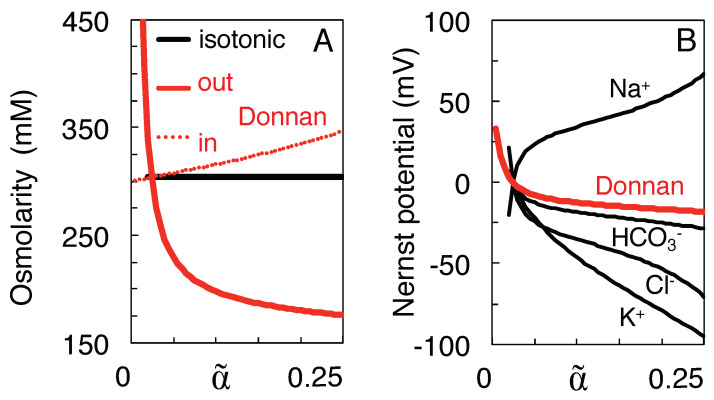
Comparison of the isotonic mixing path (black curves) and the Donnan equilibrium at each intermediate volume (red). (**A**) Extra- and intracellular total solute concentration. (**B**) Transmembrane Nernst potential. For the Donnan equilibria, the four Nernst potentials coincide at their common Donnan potential. In this and the following figures, mixing starts from the physiological concentrations of Table 1 (at $\tilde{\alpha }=\alpha =0.25$), and the concomitant shrinkage of extracellular space is monitored by variable $\tilde{\alpha }$ on the horizontal axis.

**Figure 4 membranes-13-00206-f004:**
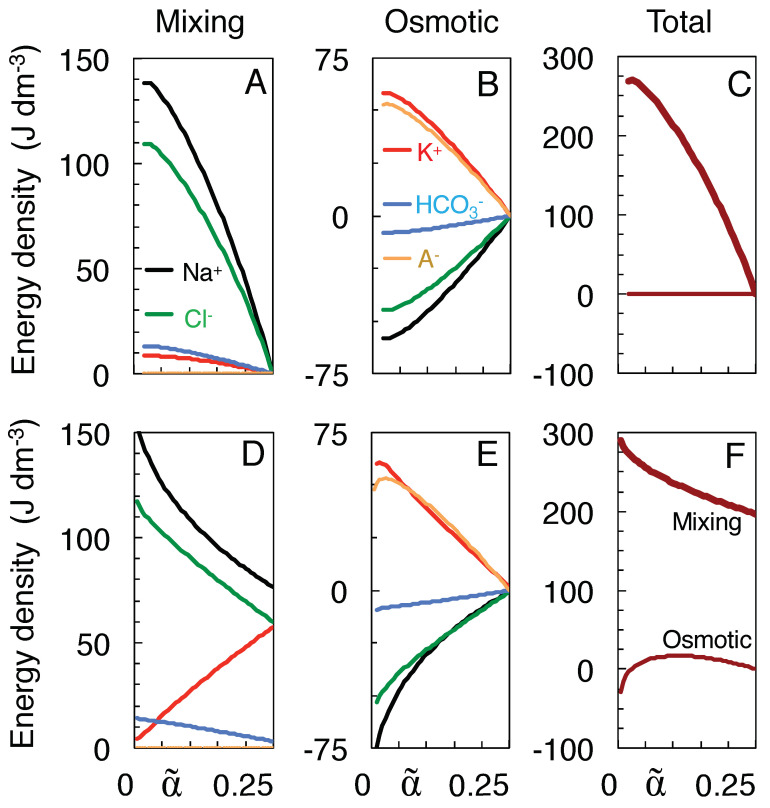
Work done by mixing (**A**,**D**), osmotic work done (**B**,**E**), and total work done (**C**,**F**). Work, expressed as energy density in joule per liter, is compared for neuronal swelling along the isotonic path (**A**–**C**) and swelling to the Donnan equilibria at the same volumes (**D**–**F**), each time starting from $\tilde{\alpha }$ = 0.25. (**A**,**D**) Work calculated from Equation (Equation 12). (**B**,**E**) Work calculated by adding Equations ([Disp-formula FD13a-membranes-13-00206]) and ([Disp-formula FD13b-membranes-13-00206]). (**C**,**F**) Total work of mixing (thick line) and total osmotic work (thin), obtained by adding over all ion species.

**Figure 5 membranes-13-00206-f005:**
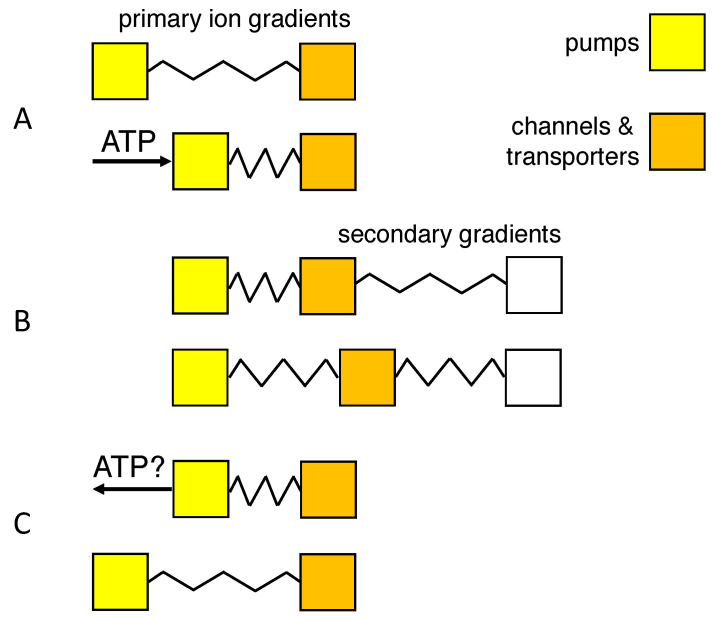
Mechanical equivalent of work of mixing. Boxes represent families of membrane proteins (yellow: the Na${}^{+}$/K${}^{+}$ pump; orange, ion channels and transporters); displacement to the right indicates work done by transport. Springs reflect storage of potential energy in primary or secondary transmembrane gradients. (**A**) The chemical potential of ATP and its metabolites drives the Na${}^{+}$/K${}^{+}$ pump and stores potential energy in the concentration gradients of the monovalent ions. (**B**) The primary ion gradients drive secondary transport of ions, water and metabolites. (**C**) Hypothetical ATP synthesis through backward operation of the Na${}^{+}$/K${}^{+}$ pump.

**Table 2 membranes-13-00206-t002:** Extra- and intraneuronal ion concentrations for the Donnan equilibrium at α = 0.25.

Ion Species	Outside (mM)	Inside (mM)
${\mathrm{Na}}^{+}$	26.6	52.5
$K^{+}$	61.2	120.9
${\mathrm{Cl}}^{-}$	54.4	27.5
${\mathrm{HCO}}_{3}^{-}$	23.4	11.9
$A^{-}$	10	134
Total	175.6	346.8

## Data Availability

Not applicable.

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
