# Peer review of "An Isotonic Model of Neuron Swelling Based on Co-Transport of Salt and Water"

_membranes, 2023, doi:10.3390/membranes13020206_

Round 1
Reviewer 1 Report
Reinoud Maex presents a very intersting article about an isotonic model of neuronal swelling due to the transport of ions and water. I only have some minor points that shall be considered before publication:
p.5 line 163 and line 265: KCC2 is not a channel.
p. 7 line 197: “Evolution of the ion concentrations during neuronal swelling”. Please re-think about the term evolution. I think, that this term is misleading. You are not talking about evolution.
Author Response
Reviewer 1
p.5 line 163 and line 265: KCC2 is not a channel.
p. 7
Reply:
'Channel' has been replaced by 'co-transporter' on lines 170 and 281.
--------------------------
line 197: “Evolution of the ion concentrations during neuronal swelling”. Please re-think about the term evolution. I think, that this term is misleading. You are not talking about evolution.
Reply:
The title now reads 'Variation of ion concentration with extracellular volume' (line 207).
Reviewer 2 Report
The authors presents an isotonic model of neuronal swelling based on co-transport of water and ions.
I enjoyed reading the paper. It is written very well, and the choices are clearly motivated.
I only have a few suggestions/comments
1. Are additional arguments possible to argue that water and salt are essentially co-tranported, e.g. the fact that the cell membrane can hardly withstand any osmotic pressure? And IF aquaporins would be present, couldn't it be that osmotic gradients are still very small if not essentially zero?
2. Is is possible to add an extra graph showing the time course towards reaching equillibrium? This could perhaps complement figure 2. From clinical and experimental observations, we know that cell swelling may take hours (or sometimes even longer); are such estimates possible, too. I understand if it is not possible; then author could perhaps spend some words in the discussion on this.
3. Will the author consider to make the code used for the simulations publicly available?
4. Is any experimental data available that can support the isotonic path? Plant cells will not follow it (if given sufficient water).
Author Response
Reviewer 2
1. Are additional arguments possible to argue that water and salt are essentially co-tranported, e.g. the fact that the cell membrane can hardly withstand any osmotic pressure?
Reply:
Here the literature on neurons is inconclusive. Andrew et al. 2007 demonstrated that pyramidal neurons can withstand enormous osmotic stresses (100 mM), whereas Murphy et al. 2017 saw no differences between neurons and glial cells (on glial cells there is a consensus that they swell under immediately osmotic stress). Neurons would, however, swell during anoxia or under stimulation with a K+ solution.
--------------------------
And IF aquaporins would be present, couldn't it be that osmotic gradients are still very small if not essentially zero?
Reply:
Some experimentalists do indeed argue that there is no osmotic stress in cytotoxic oedema. They are now cited on line 320.
For the influx of water through co-transporters no osmotic gradient is needed, as the energy is partly provided by the ion gradients. Water can even be transported against an osmotic gradient, as mentioned on line 82 (this phenomenon has been discussed in greater detail, for instance, by Zeuthen and Stein 1994).
The reviewer is right that if each infinitesimal imbalance in osmotic pressure would be immediately corrected by inflow of water through the lipid-bilayer membrane, then this would also constitute a valid implementation of isotonic swelling. This is now discussed on lines 348-351.
--------------------------
2. Is it possible to add an extra graph showing the time course towards reaching equillibrium? This could perhaps complement figure 2. From clinical and experimental observations, we know that cell swelling may take hours (or sometimes even longer); are such estimates possible, too. I understand if it is not possible; then author could perhaps spend some words in the discussion on this.
Reply:
The model is in essence based on equilibrium thermodynamics, and has no time-scale. The valid time span of the model is now discussed on lines 333-343, and may depend on the setup of the experiment or simulation.
.
--------------------------
3. Will the author consider to make the code used for the simulations publicly available?
Reply:
This is all analytical work (no simulations). Most of the formulas had been derived explicitly in reference 4.
The curves of Figure 3 plot equations 10, those of Figure 4 equations 12-13.
--------------------------
4. Is any experimental data available that can support the isotonic path?
Reply:
In Figure 3 experimental data from the literature have been added. In panels A, C and E the triangles are data compiled by Dreier from the literature (Table S2 of Dreier et al. 2013). Other data included in panel E are those of Windmueller (2005), and two studies suggested by another reviewer (Mori et al. 2002 , and Stiefel and Marmarou 2002).
I did not try to fit the model parameters to these data, even though I am confident the fit in panels A and C can be improved by enhancing the extracellular concentration of impermeant anions, which would shift the equilibrium point to the right.
--------------------------
Plant cells will not follow it (if given sufficient water).
Reply:
I suppose plant cells need the osmotic pressure (turgor) for structural stability, so this would exclude an isotonic model for them.
Reviewer 3 Report
The manuscript proposes a theoretical model describing the isotonic swelling of neurons and electro-neutral mixing of ions towards a state of full equilibrium at which the concentrations of both the impermeant and permeant ions fully equalize between the extra- and intracellular compartments. This process would be in principle reversible, giving the opportunity to neurons to counteract the energy depletion and restore ATP production thanks to a reverse activity of the Na+/K+ ATPase and some cotransporters, such as KCCs.
While the model appears to have a sound mathematical basis, concerns may be raised about the novelty of information raised here about brain physiology. The thermodynamic model sounds correct since the absence of an energy supply, in particular conditions (oxygen deprivation/cytotoxic edema), would bring the system to the lowest energy status. This implies the dissipation of ionic gradients and the osmotic swelling due to the Donnan effect. On the other hand, this model is rather simplistic and it probably does not have any physiological relevance. Particularly there are two main problems: the dissipation of the gradients and the osmotic effect would dissipate the energy which is stored in the ionic gradients and it is very unlikely that energy can be stored in these conditions. Of note, some of the work described by the author (sodium and chloride) are counteracted by potassium so, in the end, is not very clear how and when the neurons would be able to use this energy. The system described here would be reversible so that neurons may use the mixing energy. I have some serious doubts about the possibility to reverse the sodium/potassium ATPase and other transporters in these conditions (see below). The literature does not support this hypothesis and it should be demonstrated. Indeed most of the path described would be also not compatible with viable neuron/neurons due to many secondary effects (excitotoxicity, calcium overload…).
The second concern is about the two-compartment model. We lack many aspects of brain anatomy and physiology. Even though I agree that we need models to draw pictures that are simpler than the complexity of the brain. I believe we are missing many essential points (some of them correctly discussed by the author in the discussion): absence of astrocytes, the neurovascular unit and the glymphatic system, modeling ion channels permeabilities in an ischemia event. There are more comprehensive published models which take into account some of these important features as well as the Gibbs-Donnan equilibrium (PMID: 27881775, 23967075).
Other major points:
1. This manuscript widely refers to a recently published paper by the same author in Phys. Rev. E. (PMID: 34781519, see line 46). I have the feeling that no new relevant information has been generated in this new manuscript. The previous paper already studied the mixing process with different strategies and also considered the contribution of glial cells to some extent. Moreover, the hypothesis of using this energy in the dissipating gradients through a reversed operation of the Na+/K+ ATPase is already formulated. Interestingly the author in table 2 of the Phys. Rev. E. paper proposes extracellular ionic concentrations which are more similar to what might happen in vivo during ischemia. Here, unfortunately, the concentrations are related to a final state which is very unlikely to be compatible with a reversible process or with a viable neuron and which has not been observed in slice experiments.
2. I don’t think that the Na+/K+ ATPase can operate in reverse mode (line 325). It is a specialized enzyme (P-type ATPase) and an ionic pump for definition and it operated unidirectionally. There is no direct evidence in the literature which supports a reverse mode. The papers mentioned in the manuscript refer to the first decade after the discovery of the Na+/K+ ATPase and do not demonstrate the author's statement (they only show correlations with intracellular ATP levels in very extreme sodium and potassium concentrations). At that time the mechanisms of ATP synthesis were not clear and the ATP synthase was discovered in the same years and nowadays we know it accounts for ATP synthesis.
3. In the present manuscript the temporal and causal relationship are not discussed or studied, this is an essential point if we want to apply this model to cytotoxic swelling/edema. In general, the influx of primary drivers like Na+ pathways (either VGSCs or NMDARs) and secondary participants like Cl- (PMID: 25910210) and water into cells during cytotoxic edema depletes these constituents from the extracellular space (PMID: 25910210, 12427333, PMID: 12134939), causing the extracellular space to shrink. This also (probably at a later stage) would drive the extravasation of osmolytes (sodium) and water. Methodological advances over the past decades, including two-photon microscopy and MRI, have led to new insights into the role of fluid in the perivascular spaces and the glymphatic system as a source of water for edema. Cytotoxic edema sets up a new gradient for Na+, now across the blood–brain barrier, between the intravascular space and the extracellular space, which acts as a driving force for transcapillary movement of edema fluid (PMID: 12134939).
4. Ionic edema (particularly at a later stage) is intimately associated with the local blood perfusion status and edema fluid is located mostly in peri-infarct regions, with minimal edema fluids in the poorly-perfused core (PMID: 3964854, 3964854, 3964854). This is in contrast with this model which does not consider the vasculature factor and would apply mostly to core regions.
Minor:
1) 35: even though in some reports neurons swell (conflicting results, see PMID: 34498208), astrocytes are osmoresponsive cells and can recover their volume and this has an impact on the CNS compartments volume. Moreover, The sources of water driving the formation of brain edema remains a topic of debate (PMID: 32001524, 17303532).
2) 74-85: I agree that the cotransport may play a role (even though the permeability coefficients for the KCC co-transport are unknown) we can’t consider the transmembrane diffusion of water through the lipidic bilayer which is a very well established fact. The permeability for lipid bilayers is between 1 x 10-4 – 0.5 x 10-3 cm/s (PMID: 1987778, PMID: 26661240) which is similar to or more than some cotransporter may carry.
3) 136: again from these sentence one reader would think that neurons do not have permeability to water because they don’t express aquaporins (see comment above).
4) 158: the references concerning the reverse operation of KCC2 are not correct since the papers do not mention this process (others may be used like PMID: 9374636). It is also unclear to me at what stage this would happen. Maybe when extracellular potassium increases? Anyway, it is usually secondary or slower than sodium influx. Especially at the beginning astroglial cells would clear the extracellular potassium thanks to the Kir 4.1.
5) 184: the membrane potential would not affect the energy calculations but has a tremendous effect on neurons pathophysiology and at the initial stages can contribute to sodium overload of neurons (activation of NMDA receptors and also calcium overload, PMID: 9506616), the reverse operation of the sodium/calcium exchanger NCX and spreading depolarization. Indeed, it is necessary to discuss it in this type of model.
6) 357: Concerning the calcium influx, even though a direct effect of calcium on cell swelling is not demonstrated (see PMID: 25910210), it is clear that ATP depletion would reduce the calcium extrusion through the calcium pump. The depolarization itself would open voltage-gated calcium channels at least at synaptic terminals. The NCX exchangers would operate in the reverse mode (high intracellular sodium and membrane depolarization) and would further increase intracellular calcium. This would cause a calcium overload which would lead eventually to cell death.

Author Response
I thank the reviewer for his generous and expert comments. Please find my reply in the attachment.

Round 2
Reviewer 3 Report
The author significantly improved the manuscript. The discussion was extensively re-written and the limitations of the present study are now clear and fairly discussed. Some new information is also generated in Figure 3, discussing data obtained from previous models published in the field. I do not have other concerns.